# Pursuing Long-Term Business Performance: Investigating the Effects of Financial and Technological Factors on Digital Adoption to Leverage SME Performance and Business Sustainability—Evidence from Indonesian SMEs in the Traditional Market

**Florentina Kurniasari \*** , **Elissa Dwi Lestari and Hendy Tannady**

Faculty of Business, Multimedia Nusantara University, Banten 15810, Indonesia; elissa.lestari@umn.ac.id (E.D.L.); hendy.tannady@umn.ac.id (H.T.)
\* Correspondence: florentina@umn.ac.id

**Abstract:** The traditional market holds a pivotal role in Indonesia's economy because it is the main contributor to national retail grocery transactions. Nevertheless, competition with modern markets and retailers makes their competitiveness lessened. From the financial side, traditional market SMEs are vulnerable to financial risks and still face obstacles in accessing financial resources due to a lack of financial literacy. In addition, the level of digitalization of traditional market SMEs technology is also still low, so digital adoption is still a challenge that must be faced. Therefore, this study was conducted to identify the influence of financial and technological factors on the level of adoption of SME technology in traditional markets and its impact on the performance and sustainability of the SME business. This research will be conducted in a quantitative cross-sectional study of 225 SMEs in the traditional market. The sampling technique that will be used is judgmental sampling. This study's research data will be analyzed using SEM-PLS. The study result shows that financial literacy positively affects SME financial accessibility and financial risk. The study also confirms that financial accessibility, performance expectancy, effort expectancy, and social influence are variables that significantly affect SMEs digital adoption, while the effect of financial risk on digital adoption is found to be insignificant. The study result also shows that SMEs digital adoption is positively affecting their performance, which eventually affects their business sustainability.

**Keywords:** business sustainability; SME business performance; technology adoption; financial factor; traditional factor

## 1. Introduction

Business sustainability is a popular topic for discussion and research among practitioners and academics [1]. According to Ye and Kulathunga [2], a sustainable business is one that expands over time and generates long-term benefits. As a result, from an economic standpoint, studies on business sustainability contribute to efforts to achieve long-term economic growth. An important sector that needs attention and focus from the government is SMEs. According to the website of the Ministry of Economic Coordination of the Republic of Indonesia, SMEs play an important role in the growth of Indonesia's economy. Of all the units of business that exist in Indonesia today, 99% are SMEs. SMEs also account for 60.5% of Indonesia's GDP and 96.9% of national labor absorption [3]. These data are an important indicator of how SMEs have become a critical engine of the Indonesian economy [4,5] and should receive important attention from various parties, especially the government [6], especially during times of crisis. During the COVID-19 outbreak, SMEs, which are often resilient and capable of surviving difficult situations (such as the

1997–98 monetary crisis), had a major downturn due to restrictions on economic and social activities that impair their business survivability [7]. Moreover, even though it was reported that 84.8% of SMEs have returned to normal operation after passing the COVID-19 pandemic [8], SMEs, as the business entity that suffered the greatest loss from COVID-19, typically lack sufficient resources, particularly financial and managerial resources, and are unprepared for such disruptions that are likely to last longer than expected [5,9]. As a result, the SMEs business sustainability must be preserved in such a way that it doesn't adversely affect the country's economic and social standing [4].

One of the most important SME sectors in the economy as a whole is the retail sector. Traditional retail businesses control 65–70% of total retail sales in Indonesia, with product supply chain value to traditional retail even estimated to reach at least US$58 billion (about Rp817 trillion) per year [10]. Traditional markets account for the vast majority of traditional retail transactions in Indonesia. Indonesia recorded wholesale retail sales of US$71.64 billion in 2021. Sales through traditional markets still dominate the value of domestic wholesale retail sales. Traditional and wholesale retail transactions totaled US$53.59 billion, or 74.8% of total national retail sales [11]. Despite having a strategic role in the Indonesian economy, in fact, traditional markets experienced negative growth of −8.1% while modern markets grew 31.4% [12]. According to the Indonesian Minister of Commerce, there are at least three factors that cause the traditional retail sector to concede its position to the modern retail industry. First, customers prefer the modern retail market over the traditional market because it is cleaner and more convenient, offers a fixed and stable price, and provides goods of good standard quality [13]. Second, on the supply chain side, the modern retail market acquires goods directly from the producer, allowing it to sell more varieties of goods at lower prices. Third, in terms of capital, modern retail market traders have more capital than small retail traders in traditional markets [14]. Moreover, during pandemic times, traditional market traders experienced a decrease in trading turnover in the range of 40–70% [15]. Although many scholars have already researched the performance of SMEs, there are still very few studies in Indonesia that investigate the performance and sustainability of SMEs in the retail industry, especially studies on SMEs of traditional traders. Therefore, the study of the performance and sustainability of the businesses of traditional market traders offers a novelty in research that has not been much studied before by academics.

Extensive literature analyses have revealed that enabling digital technology and digitalization have become organizational innovation enablers because they support value-creating, value-delivery, and value-capture opportunities in business contexts and are critical to guaranteeing business sustainability [16–18]. As a result, SMEs capacity for digital adoption also affects their ability to establish business sustainability. The rapid development of technology plays a major role in transforming the working processes of every aspect of the business [5,19], starting from enabling better access to financial resources [2], promoting better marketing processes, and making supply chain operations more efficient [20,21] for SMEs. Based on the report released by We Are Social and KEPIOS, the number of Indonesians who access the internet continues to increase. 196.7 million users by 2020; 202.6 million users in 2021; and 210 million users in 2022) [20]. There are 100 million Indonesians who own more than one mobile phone. According to the survey conducted by the Association of Internet Service Providers Indonesia (APJII) in 2022, the rate of active participation in internet use in Q1 2022 increased by 77.02% compared to Q1 in 2021. This percentage of active penetration or participation also increased from quarter I in 2020 to quarter 1 in 2021, which was 73.7%. Despite the rapid rise of technology among SMEs, the rate of technology adoption has remained largely low [6], especially in Indonesia. Only 15 of the 38 provinces in Indonesia have a 100% percentage of SME participants who use the internet in their business operations. In fact, there are two provinces where internet use for business is still below 35%. These two provinces are Aceh (31.25%) and East Nusa Tenggara (20%). In terms of the use of accounts on the marketplace to market products and services, only 22% of SMEs have already opened accounts there. Based on website

ownership to support a business, 75.49% still do not have a website. Still, 17% of SMEs do not use social media to sell products or services [22].

According to research conducted by Gomes et al. [23] on several industrial sectors in Brazil, an organization with good performance will be able to maintain business sustainability, not only in the manufacturing industry but also in the service industry. Still, according to Gomes et al. [23], if an organization is able to realize an excellence management system and produce products that meet the criteria of the global market, then this will ensure the sustainability of the business of the industry. Basically, individuals and organizations will choose to adopt technology and digitalization when assessed as being able to increase competitive value in the market [24]. Technological innovation has a significant impact on employee performance, which will imply organizational performance. Digital adoption will help reduce human errors, improve the efficiency and effectiveness of work, and increase speed in terms of coordination and communication [24]. Business transparency and efficiency will increase as management uses technology within an organization [24,25]. Digital adoption will also force management to create new policies and strategies that will result in new directions for leadership, improve workforce efficiency, strengthen industry competitiveness, and provide overall benefits to organizations [26,27], as well as provide overall benefits to organizations [28]. However, the adoption of digitalization also requires maximum effort from management, employees, and the organization as a whole [29].

Regarding the impact of financial aspects on digital adoption, the organization's access to funding sources will play an important role in supporting its innovation capacity, one of which is digital adaptation [30]. Financial technology as a sharing economy platform could play an important role in helping SMEs received better financial access to maintain their performance and sustainability [31]. The better access organizations have to credit sources and investors, the more digital adoption will increase and impact efficiency and innovation [32]. Digital adoption requires companies to have a residential funding resource. Organizations with limited or minimal access to funding will potentially have a low probability of innovation and digital adoption. Barriers to financial access are significant in limiting the availability of digital adoption [33]. In practice, innovation is not free, so barriers to financial accessibility will lead to the failure of digital adoption [32]. Companies with more funding tend to become more flexible and have greater capacity to invest in research and development, which will lead to better rates of success in the use of technology and digitalization [34]. In addition to financial accessibility, another factor that can interfere with the adoption of digitalization for SMEs is risk. The form of risk assessed as essential in decision-making in the business sector is financial risk [35,36]. Knowledge and awareness of financial risks will make management reluctant to adopt technology and digitalization [36,37]. Previous research has also shown that financial risks have a negative impact on digital adoption [38,39]. To equip SME participants in terms of maturity of knowledge, information, and understanding of financial accessibility and financial risk, financial literacy is also required.

In addition to financial aspects, other aspects that are closely related to digital are technological factors. One technological factor that influences digital adoption is performance expectancy. A study conducted by Andrews et al. [40] that investigated the intention to adopt AI technology and digitization devices among librarians concluded that performance expectancy had a positive impact on increasing digital adoption. Another study conducted by Majeed et al. [41] that investigated consumer behavior in the United States and China related to the adoption of travel booking applications concluded that performance expectancy positively affects application adoption. Other studies support the impact of performance expectancy on digital adoption [42–44]. When a user determines which technology or tool to use by placing ease of use as a primary priority, this is called an "effort expectation" [45]. Based on the behavior of technology use, generally, consumers will definitely choose technology or applications that make their lives more practical, as will employees in the organization [46]. A study conducted by ALraja [44] that tested the impact of effort expectancy on digital adoption, in particular e-government adoption in Oman,

demonstrated that effort expectancy had a significant impact on e-government adoption. Other research discussing the relationship between the two also confirms that there is a significant influence of effort expectancy on technology adoption [44,47,48]. Another technological factor that contributes to digital adoption is social influence. Consumers have characteristics that tend to make them pay attention to and follow the lifestyle and trend developments that occur around them, including in the process of business operations. When a lot of society or SME actors use digitalization to support their businesses, of course, this will be a stimulus for other SMEs to also use digitalization [49]. Research results show that social impact or influence will curtail and force digital adoption [50,51].

Even though many studies have explored the digital adoption of SME, there are limited studies that relate the digital adoption to SME performance and business sustainability in the context of the traditional market. This study is expected to fill all of the aforementioned gaps regarding the factors that drive FinTech digital adoption and how it affects SME business performance and sustainability, particularly for traditional market traders in developing nations, particularly in the context of Indonesia. Furthermore, this study aims to test and analyze the impact of financial aspects (that consist of financial literacy, financial accessibility, and financial risk) and UTAUT's technological factors (performance expectancy, effort expectancy, and social influence) on digital adoption, as well as their implications for SME performance and SME sustainability in the Regency of South Tangerang, Indonesia.

The 225 data points were gathered through empirical and personal survey questionnaires distributed to traditional SME traders in Tangerang Regency, Indonesia. The Partial Least Squares-Structural Equation Modeling (PLS-SEM) in Smart PLS 3 Version 4 is utilized for hypothesis testing and analysis in this study. In specific, the research questions posed by this study are as follows:

Q1. Whether financial literacy significantly affects SMEs financial accessibility and financial risk?
Q2. Whether financial accessibility and financial risk affect FinTech's digital adoption?
Q3. Whether performance expectancy, effort expectancy, and social influence affect FinTech digital adoption?
Q4. Whether FinTech digital adoption affects SME performance?
Q5. Whether FinTech digital adoption affects SME sustainability?

This study also contributes to strong SMEs business performances and sustainability by focusing on how financial technology adoption will lead to sustainable finance and innovation.

## 2. Literature Review

### 2.1. Resource-Based View

Resource-Based View (RBV) theory is a theory explaining the importance of an organization's own internal resources and capabilities in order to win the competition [52]. As a response to the advancement of information and communication technology (ICT), organizations have to acquire technology skills as well as invest in developing some technology platforms [50]. The new adoption of ICT-related strategies puts traders in the traditional market one step ahead of their competitors. It is more efficient.

### 2.2. The Unified Theory of Acceptance and Use of Technology

As a theoretical framework, UTAUT (Unified Theory of Acceptance and Use of Technology) theory is widely used to analyze individual behavior in adopting new technology [45]. The UTAUT theory explained the main determinants that affect people's intentions to adopt new technology, namely: performance expectancy, effort expectancy, social influence, and facilitating conditions. The UTAUT theory is extended with the additional explanation of some variables to moderate the relationship between these factors: gender, age, experience, and voluntariness [53]. In the study conducted by Kurniasari et al. [54], the variables of performance expectancy, effort expectancy, and social influence were proven to have a significant effect on adopting the new technology platform in the financial industries.

### 2.3. Financial Literacy and Financial Accessibility

Financial literacy is usually defined as the individual level of financial knowledge that is needed to make financial decisions [55]. Financial literacy has been considered a crucial tool for the success of SMEs since it assists in understanding and evaluating the data needed to make day-to-day financial decisions in the firm's day-to-day management [56]. Adequate financial knowledge will lead to better skills and capabilities related to financial issues. Traditional market SMEs that acquire financial knowledge have more confidence to run their businesses better [7]. Having high financial literacy would help traditional market traders build their relationships with financial institutions. The study developed by Okello et al. [57] found that SMEs need to improve their financial literacy to obtain more financial access from financial institutions [2]. SMEs with good financial literacy are able to manage their financial transactions, including setting up budgets, controlling costs, and preparing the financial report that is needed to obtain loans from external financial sources. Based on these findings, this study proposes the first hypothesis:

**H1.** *Financial literacy positively affects financial accessibility.*

### 2.4. Financial Literacy and Financial Risk

Every SME should be mindful of the volatility in today's unstable economic environment, as it might have an impact on their business performance. SMEs should strengthen their financial capabilities by updating their financial knowledge [58]. Financial knowledge is important in managing the potential risk [2]. The study by Menike [56] mentioned that better financial literacy will lead to better risk tolerance before making crucial financial decisions. The financial decision needs to be more specific based on the available trusted information, with the objective of profit maximization or cost efficiency. Based on these arguments, this research proposes the second hypothesis:

**H2.** *Financial literacy positively affects financial risk attitude.*

### 2.5. Financial Accessibility and Digital Adoption

According to a literature review, legal institutions determine access to external capital and business growth. In other words, in countries with better institutional systems, funding impediments are fewer, enterprises acquire more external finance, and they may grow faster [58]. External finance availability for small and medium-sized businesses (SMEs) is a topic of significant research attention among academics and constitutes an important issue for policymakers globally [59,60]. SMEs, in particular, face challenges when it comes to acquiring external financing from banks and capital markets due to their size and typical opaqueness [2,61]. Studies conducted by Irwin and Scott [62] and Khan et al. [63] mentioned that financial accessibility is SMEs main constraint in growing their businesses. Financial accessibility refers to the ability of SMEs to receive financial services that fulfill their financial needs, such as loans, savings, or payments [64]. The wider financial access enables SMEs to adopt various financial digital platforms that are offered by financial institutions. On the contrary, the limited access to financial instruments will make SMEs reluctant to adopt a digital platform and prefer using internal funding [65]. According to Hau et al. [66], FinTech financing reduces supply frictions (such as a considerable geographic distance between debtors and the nearest retail bank) and helps entrepreneurs with lower credit scores acquire credit in the Chinese market. Moreover, Tang [67] discovers that, in the United States, FinTech credit supplements bank lending for small-scale loans. This argument leads to the following hypothesis:

**H3.** *Financial accessibility positively affects SMEs digital adoption.*

### 2.6. Financial Risk and Digital Adoption

In an unpredicted environment, each organization needs to have capabilities for managing both financial and non-financial risks. Financial risks are related more to the

negative financial losses that can affect the business's performance, such as credit risks, market risks, liquidity risks, and failure to use new technology. Meanwhile, non-financial risks are more closely related to negative outcomes due to non-financial issues, such as reputational risks [68]. The adoption of the new technology is expected to assist traditional traders in managing these financial risks. A risk mitigation strategy can be considered by investing some funds in expanding the business, investing in some profitable financial instruments, and preparing an amount for emergency fund purposes [2]. New technology platforms provide real-time data accuracy, speedy monitoring analysis to prevent fraud, quick responses to customer complaints, and efficient decision-making [69]. Based on these arguments, this study offers the proposed hypothesis:

**H4.** *Financial risk positively affects digital adoption.*

### 2.7. Performance Expectancy and Digital Adoption

Performance expectancy is well-related to the individual's belief that applying the new technology will increase performance [45]. The study of Tajul Urus et al. [70] found that performance expectancy has a positive influence on the adoption of digital payments among fresh graduates both in Indonesia and Malaysia. Furthermore, Ye and Kulathunga [2] explained that performance expectancy was related to the ability of the new digital platform to provide a system to perform tasks quickly that were needed to expand the business. Moreover, according to Najib, Ermawati, et al. [61], performance expectancy positively affects FinTech adoption in small food businesses. Moreover, Alshebami [71] also found that performance expectancy positively affects the adoption of e-crowdfunding platforms as the source of business financial resources. Studies by Alkhwaldi et al. [72] and Chan et al. [73] show the performance expectancy variable to be a significant driver for FinTech adoption in Jordan and Australia. Based on these arguments, this study comes up with the proposed hypothesis:

**H5.** *Performance expectancy positively affects digital adoption.*

### 2.8. Effort Expectancy and Digital Adoption

Effort expectancy is well-defined as the ease of using the new technology platform [45]. The ease is related to the simplicity of the new platform features. Traditional traders are willing to adopt the new technology if they feel there are fewer difficulties and no extra effort is needed to access it. In the context of the adoption of digital financial platforms, the study of Tajul Urus et al. [70] was able to find significant influence on the adaptation of new digital payment services. The adoption of technology will be readily accepted by its users if they feel the ease of using its features. According to Maryam et al. [74], perceived effort expectancy substantially influences FinTech adoption, and the positive relationship indicates that the greater the effort expectancy, the greater the FinTech adoption. According to Najib and Fahma [75], the simplicity of use of digital payment methods makes them more appealing to SMEs in Indonesia. Therefore, this study proposes the next hypothesis:

**H6.** *Effort expectancy positively affects digital adoption.*

### 2.9. Social Influence and Digital Adoption

Social influence is related to the level of other people's influence on individual decision-making [45]. Some studies are able to analyze the significant impact of social influence on technology adoption in some developing countries, such as Indonesia and Malaysia [54]. Indonesia is well-known for its tied-knot extended families, in which inner family circles and friends are the most trusted people in making crucial decisions [70]. According to Najib, Ermawati, et al. [61], the strongest determinants of FinTech adoption among small food business owners in Indonesia are social influences. Based on this argument, this study proposes the following hypothesis:

**H7.** *Social influence positively affects digital adoption.*

## *2.10. Digital Adoption and SMEs Performance*

One of the most important characteristics that distinguishes successful from failed organizations is their ability to identify and capitalize on technology opportunities [17]. Several studies demonstrate that SMEs capabilities in adopting the new digital technology platform are significantly related to their business performance [7,76]. Huston [77] also found that financial digital adoption platforms are able to cope with the advancement of information technology. For SMEs, digital adoption will allow them to receive proposed financing and loans from financial technology providers since it's easy to download the FinTech application [4]. Digital adoption will lead to more efficient business processes, simplified documentation, and speedy financial decision-making. The research conducted by Ye and Kulathunga [2] found that the digital adoption of financial technology will enable SMEs to reduce operating costs and increase business performance. In addition, the business's performance can be proven by growing market shares, increasing profit, having quicker response times, a quicker return on investment, and reducing customer complaints [78]. Therefore, this study comes up with the proposed hypothesis:

**H8.** *Digital adoption positively affects SMEs performance.*

## *2.11. SMEs Performance and SMEs Sustainability*

The study of Widagdo and Sa'diyah [7] found that SMEs with outstanding business performances will be able to adapt to the competitive environment and maintain their sustainable performances in the long-run [79,80]. The importance of access to finance for the sustainability of SMEs is widely discussed in the previous literature. Huston [77] showed that SMEs sustainability is strongly related to their business performance. Accessibility to the source of funding increases SMEs business performances, especially when expanding their market growth and strengthening their capital. The SMEs sustainability is shown by the expansion of the market, production capacity improvement, strong business competitiveness and capabilities [4], and better management skills [56]. Therefore, the following hypothesis is proposed:

**H9.** *SMEs performance positively affects SMEs sustainability.*

Figure 1 is the proposed research framework:

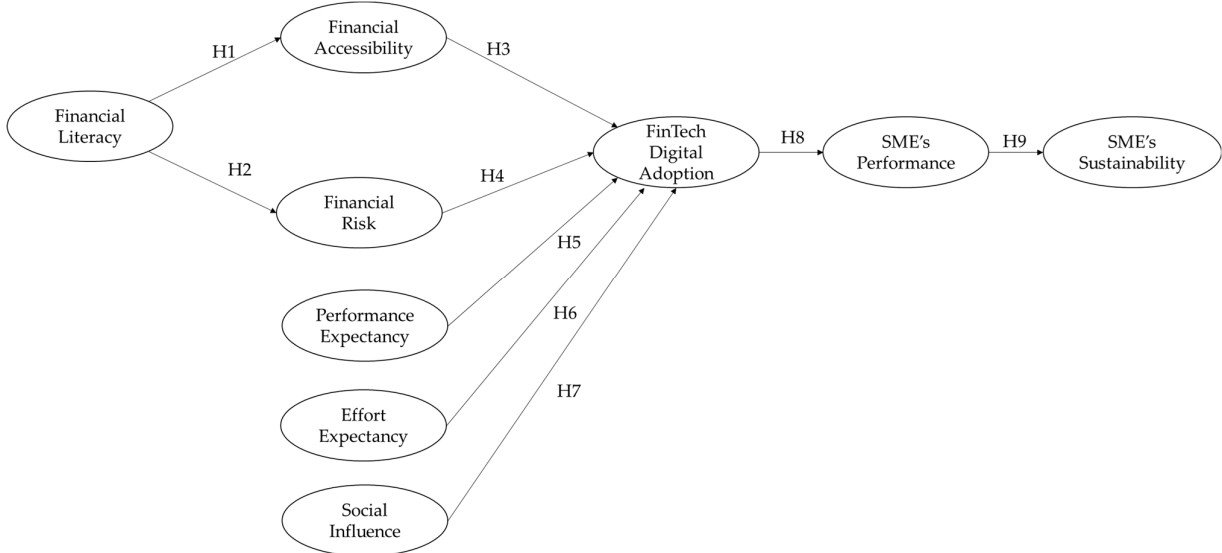

**Figure 1.** Research framework.

## 3. Methodology

### 3.1. Procedure and Participants

This study used a cross-sectional design in which data were gathered at a specific point in time [81]. The primary data for the study were collected using a self-administered electronic questionnaire. Respondents were asked to provide feedback on several questions in the questionnaire during the survey. The survey used multi-item scales adapted from previously validated studies to assess the effects of financial literacy, financial accessibility, financial risk, performance expectancy, effort expectancy, social influence, digital adoption, business performance, and sustainability on Tangerang city traditional market traders.

### 3.2. Sampling and Measures

In this study, we used non-probability sampling with a judgmental technique. The respondents for the current study were chosen based on the following characteristics: (1) SMEs with an owner; (2) SMEs that have been in business for more than three years and are still in business; (3) SMEs that have been using financial technology to support their business; and (4) SMEs that operate in a traditional market.

Respondents' perceptions have been measured using a self-administered, validated questionnaire. All items have been evaluated on a five-point Likert scale, with scores ranging from 1 (strongly disagree) to 5 (strongly agree). The five-point Likert scale is useful in measuring the variables of user characteristics and comparing the reliability coefficients among variables. A five-point scale rather than a seven-point scale was chosen for a number of reasons, one of which was that it became possible to compare reliability.

All the question indicators used to measure the variables in this study are taken from the results of previous studies that have proven to have good levels of validity and reliability, whereas the question indicator for financial literacy is taken from research conducted by Okello et al. [82]. As for the indicator question for the variable access to finance, it was taken from the study conducted by Ye and Kulathunga [2], and the question for the financial risk attitude variable was taken from the research carried out by Blais and Weber [83]. In this study, the determinant factor of financial technology user acceptance is adapted from the UATUT framework (effort expectancy, performance expectancy, and social influence) by Venkatesh et al. [45], which has been modified to fit with the research context. In particular, the measurements of effort expectancy, performance expectancy, and social influence variables in this study are based on Najib, Ermawati, et al. [61] and questions that have been proven valid and reliable in measuring FinTech adoption by food SMEs in Indonesia. While the questions for SME performance variables are taken from Hudson et al. [78], the question indicators for SME sustainability variables are taken from research conducted by Ye and Kulathunga [2].

The sample size was calculated using G-Power version 3.1. Based on a power of 0.95 (which should be higher than 0.80 in behavioral and social science research) and an effect size of 0.15, a sample size of 151 was required to test the model with eight predictors. PLS-SEM requires a minimum sample size of 100 [84]. As a result, 225 samples for this study were collected.

### 3.3. Statistical Data Analysis

The questionnaire survey data sets were analyzed, and all proposed relationships were investigated using the partial least squares method, a second-generation multivariate technique. Structural Equation Modelling (SEM) is used as a multivariate statistical analysis method to measure both measurement and structural model as well as minimize the error variance [85,86]. The research framework was analyzed using SmartPLS version 4, which is able to measure the relationship among variables with a smaller sample size with no error in the value of the indicator variable. Since the sample size of the study is only 225, the Bootstrapping mechanism (5000 resamples) was applied to determine the path's significance level [83].

Harman's single factor test was used in the Statistical Package for Social Sciences to ensure the absence of common method bias in the questionnaire. This test analysis showed that the first factor only accounted for 37.924% of the variance, a percentage that falls short of the given cutoff point of 50% of the total variance [87]. As a result, there is no common method bias observed.

## 4. Results

### 4.1. Descriptive Analysis Results

The demographic profile of respondents was examined using a descriptive analysis of the study. According to Table 1, the majority of market traders who responded to this survey are male, or 65.33%; aged 36 to 45 years, or 46.22%; diploma and high school graduates, or 63.55%; and have a daily income of Rp 500,000 to Rp 1,000,000. This study also examines the reasons for FinTech usage among the respondents. The majority of respondents used FinTech because it was easy to use and handled business transactions quickly.

Table 1 below shows the descriptive analysis of the respondents in each category.

**Table 1.** Results of Descriptive Analysis.

|  | Category Answers | Numbers | (%) |
|---|---|---|---|
| Gender | Male | 147 | 65.33% |
|  | Female | 78 | 34.67% |
| Age | 17–25 years old | 10 | 4.44% |
|  | 26 to 35 years old | 96 | 42.67% |
|  | 36 to 45 years old | 104 | 46.22% |
|  | Above 46 years up | 15 | 6.67% |
| Education Level | Elementary School Graduate | 5 | 2.22% |
|  | Junior High School Graduate | 33 | 14.67% |
|  | High School Graduate | 68 | 30.22% |
|  | Graduate Diploma | 75 | 33.33% |
|  | Bachelor Graduate | 42 | 18.67% |
|  | Postgraduate | 2 | 0.89% |
| Daily Business Income | Less than 500,000 Rupiah | 22 | 9.78% |
|  | 500,000–1,000,000 million Rupiah | 103 | 45.78% |
|  | 1–2.5 million Rupiah | 68 | 30.22% |
|  | 2.5 million–Rp 5 million Rupiah | 24 | 10.67% |
|  | More than 5 million Rupiah | 8 | 3.56% |
| Reason to use FinTech | Easy to use | 101 | Multiple responses |
|  | Quick speed | 78 |  |
|  | Safe and secure | 34 |  |
|  | Affordable | 16 |  |

### 4.2. Analysis of Measurement (Outer Model)

A first-order reflective construct was used to conceptualize all of the exogenous and endogenous variables in the study. The presentation of measurement model results is an important aspect of PLS model evaluation because it focuses on determining item reliability, internal consistency reliability, convergent validity, and discriminant validity of the indicators used to depict each construct [84–86,88].

Table 2 depicts the assessment of the measurement model. The convergence validity of the items was assessed by outer loadings and average variance extraction (AVE). The outer loading analysis is driven by the theoretical relationships among the observed and unobserved variables [89]. The indicator reliability is assessed by examining each single indicator loading in the study to determine whether the constructs meet the 0.708 criterion requirements [85]. All of the single indicator loadings in the study are above the 0.708 threshold except for four items (EFF_EXPC_2, PERF_EXPC_2, PERFORMANCE_1, PERFORMANCE_3), which, for research purposes, are not used in the process of further

analysis. This, therefore, provided support for convergent validity (see Ref. [90]). The AVE values of 0.616 to 0.808 are well above the minimum required level of 0.50 [91], thus also demonstrating the convergent validity of all constructs. The reliability of each item was assessed by calculating Cronbach's alpha (CA) and composite reliability (CR). The reliability measures in this study are above the acceptable satisfactory levels (Cronbach's alpha > 0.70, composite reliability > 0.70) as recommended by scholars Blais and Weber [83], Hair et al. [86,90].

**Table 2.** Indicator Reliability, Reliability, and Convergent Validity.

| Latent Variable | Observed Variable | Outer Loadings | Alpha | CR | AVE |
|---|---|---|---|---|---|
| Access to Finance | ACC_FIN_1 | 0.760 | 0.904 | 0.905 | 0.635 |
| | ACC_FIN_2 | 0.790 | | | |
| | ACC_FIN_3 | 0.810 | | | |
| | ACC_FIN_4 | 0.810 | | | |
| | ACC_FIN_5 | 0.811 | | | |
| | ACC_FIN_6 | 0.794 | | | |
| | ACC_FIN_7 | 0.799 | | | |
| Financial literacy | FIN_LIT_1 | 0.746 | 0.875 | 0.878 | 0.616 |
| | FIN_LIT_2 | 0.793 | | | |
| | FIN_LIT_3 | 0.814 | | | |
| | FIN_LIT_4 | 0.798 | | | |
| | FIN_LIT_5 | 0.749 | | | |
| | FIN_LIT_6 | 0.807 | | | |
| Financial risk | FIN_RISK_1 | 0.936 | 0.923 | 0.990 | 0.808 |
| | FIN_RISK_2 | 0.855 | | | |
| | FIN_RISK_3 | 0.935 | | | |
| | FIN_RISK_4 | 0.866 | | | |
| Effort expectation | EFF_EXPC_1 | 0.881 | 0.721 | 0.722 | 0.782 |
| | EFF_EXPC_3 | 0.888 | | | |
| Performance expectation | PERF_EXPC_1 | 0.882 | 0.738 | 0.741 | 0.792 |
| | PERF_EXPC_3 | 0.898 | | | |
| Social influence | SOC_INFL_1 | 0.855 | 0.816 | 0.816 | 0.731 |
| | SOC_INFL_2 | 0.850 | | | |
| | SOC_INFL_3 | 0.861 | | | |
| Digital adoption | DIG_ADPT_1 | 0.844 | 0.790 | 0.794 | 0.704 |
| | DIG_ADPT_2 | 0.810 | | | |
| | DIG_ADPT_3 | 0.863 | | | |
| SME performance | PERFORMANCE_2 | 0.797 | 0.906 | 0.906 | 0.681 |
| | PERFORMANCE_4 | 0.812 | | | |
| | PERFORMANCE_5 | 0.821 | | | |
| | PERFORMANCE_6 | 0.836 | | | |
| | PERFORMANCE_7 | 0.832 | | | |
| | PERFORMANCE_8 | 0.852 | | | |
| SME sustainability | SUSTAINABILITY_1 | 0.827 | 0.795 | 0.797 | 0.709 |
| | SUSTAINABILITY_2 | 0.829 | | | |
| | SUSTAINABILITY_3 | 0.870 | | | |

The Fornell-Larcker criterion, the HTMT ratio, is used to validate discriminant validity. According to the Fornell-Larcker criterion [91], the square root of AVE should be greater than the correlation between the construct and the other constructs. According to Table 3, the square roots of the AVE values measured on the constructs exceed the correlations shared by the constructs and others. In other words, constructs are self-contained. Because the model meets the criteria, it has good discriminant validity.

**Table 3.** Discriminant Validity—Fornell-Larcker criterion.

|  | 1 | 2 | 3 | 4 | 5 | 6 | 7 | 8 | 9 |
|---|---|---|---|---|---|---|---|---|---|
| 1. Financial Access | 0.797 |  |  |  |  |  |  |  |  |
| 2. Digital Adoption | 0.647 | 0.839 |  |  |  |  |  |  |  |
| 3. Effort Expectancy | 0.595 | 0.653 | 0.884 |  |  |  |  |  |  |
| 4. Financial Literacy | 0.727 | 0.622 | 0.584 | 0.785 |  |  |  |  |  |
| 5. Financial Risk | 0.156 | 0.023 | −0.028 | 0.142 | 0.899 |  |  |  |  |
| 6. SMEs Performance | 0.574 | 0.698 | 0.603 | 0.580 | −0.115 | 0.825 |  |  |  |
| 7. Performance Expectancy | 0.659 | 0.636 | 0.639 | 0.579 | 0.028 | 0.582 | 0.890 |  |  |
| 8. Social Influence | 0.572 | 0.627 | 0.593 | 0.485 | −0.066 | 0.621 | 0.624 | 0.855 |  |
| 9. SMEs' sustainability | 0.555 | 0.689 | 0.610 | 0.576 | −0.120 | 0.765 | 0.570 | 0.554 | 0.842 |

The Fornell-Larcker approach has been criticized in current methods for calculating discriminant validity for a variety of research questions. As a result, the current study also used a multitrait-multimethod matrix to calculate discriminant validity, also known as the heterotrait-monotrait correlation ratio [88]. Table 4 shows the HTMT ratio to find the discriminant validity. Hair et al. [88] declared that the "Heterotrait-Monotrait ratio" should be "less than 0.90" to confirm variable discriminant validity. Using this criterion, it was discovered that all of the values were less than 0.90, indicating that there was no problem with discriminant validity.

**Table 4.** Discriminant Validity—HTMT.

|  | 1 | 2 | 3 | 4 | 5 | 6 | 7 | 8 | 9 |
|---|---|---|---|---|---|---|---|---|---|
| 1. Financial Access |  |  |  |  |  |  |  |  |  |
| 2. Digital Adoption | 0.764 |  |  |  |  |  |  |  |  |
| 3. Effort Expectancy | 0.735 | 0.864 |  |  |  |  |  |  |  |
| 4. Financial Literacy | 0.814 | 0.746 | 0.734 |  |  |  |  |  |  |
| 5. Financial Risk | 0.158 | 0.066 | 0.071 | 0.144 |  |  |  |  |  |
| 6. SMEs Performance | 0.633 | 0.823 | 0.746 | 0.648 | 0.141 |  |  |  |  |
| 7. Performance Expectancy | 0.804 | 0.828 | 0.875 | 0.721 | 0.082 | 0.713 |  |  |  |
| 8. Social Influence | 0.665 | 0.779 | 0.772 | 0.573 | 0.110 | 0.722 | 0.803 |  |  |
| 9. SMEs sustainability | 0.655 | 0.867 | 0.807 | 0.687 | 0.153 | 0.899 | 0.748 | 0.687 |  |

### 4.3. Structural Analysis (Inner Model)

In partial least squares (PLS) analysis, the $R^2$ value, which evaluates the coefficient of determination, and the level of significance of path coefficients are the primary evaluation criteria for the goodness of the structural model [85,86,92]. The $R^2$ values range from 0 to 1, with higher values indicating better explanation power. In several social science areas, $R^2$ values of 0.75, 0.50, and 0.25 are regarded as significant, moderate, and weak, respectively [88,91]. Based on Table 4, $R^2$ for financial accessibility is 0.529 and financial risk is 0.016. In other words, variable financial literacy can explain financial accessibility at 52.9% and financial risk at 1.6%. $R^2$ for the digital adoption variable is 0.571. In other words, 57.1% of digital adoption can be explained by variables such as financial accessibility, financial risk, performance expectancy, effort expectancy, and social influence. Furthermore, $R^2$ for SMEs' performance is 0.485, and $R^2$ for SMEs sustainability is 0.583. The $R^2$ value indicates that SME performance variables can be explained by the digital adoption variable of 48.5% and the SME business sustainability variable can be described by the SMEs performance variability of 58.3%. Based on the effect size invented by Henseler et al. [93],

it can be concluded that the effect size $R^2$ for financial risk is weak, while $R^2$ values for variables such as financial accessibility, digital adoption, SME performance, and SME business sustainability are significant.

Hypothesis Testing

The bootstrapping approach suggested by Blais and Weber [83], Purwanto, and Sudargini [94] was used to assess the significance of the path coefficients [85]. As a result, we used 5000 subsamples to evaluate the significance of path coefficients at the 5% level of significance. The study tested hypotheses by assessing the values of β and T Statistics. The value of β specifies the expected variation in a construct that is dependent on the variation of units in another construct. The greater the β value, the more significant the effect on endogenous latent construction. However, its significance must be confirmed using a T-statistical test [95]. Furthermore, in accordance with Hair et al. [85], this study evaluated the effect size (f2) to demonstrate substantive significance. Statistical significance, such as the *p* value, can only indicate whether an effect exists but not the magnitude of the effect. As a result, both statistical significance (*p* value) and substantive significance (f2 value) measurements are required for accurate data reporting and interpretation [96]. Meanwhile, the size of the effect of f2 in this study refers to Henseler et al. [93], who stated that 0.02, 0.15, and 0.35 represent the measurement of small, medium, and substantial effects, respectively. According to Table 4, this study accepts eight of the nine hyphotheses.

In this research, we tested nine direct hypotheses. Based on Table 5, financial literacy positively affects SME financial accessibility (β = 0.727, T value = 17.289, *p* value = 0.000, with a substantial effect where and SMEs financial risk (β = 0.142, T value = 2.217, *p* value = 0.013, with a small effect where f2 = 0.021). Thus, Hypotheses 1 and 2 are supported. This study's third and fourth hypotheses were established to assess the direct effect of financial accessibility and financial risk on SME technology adoption. Based on Table 4, it is only financial accessibility that positively affects SME digital adoption (β = 0.256, T value = 3.413, *p* value = 0.000, with a small effect where f2 = 0.073), while the financial risk effect on SME digital adoption is insignificant (β = 0.001, T value = 0.014, *p* value = 0.494, with no effect). Therefore, Hypothesis 3 is supported, while Hypothesis 4 is unsupported. Hypotheses 6, 7, and 8 of this research were postulated to evaluate the direct effect of performance expectancy, effort expectancy, and social influence on SMEs digital adoption. Based on Table 5, Hypotheses 6, 7, and 8 were all supported. The result shows that performance expectancy is positively affecting digital adoption (β = 0.154, T = 1.936, *p* = 0.026, with a small effect where f2 = 0.024), effort expectancy is positively affecting digital adoption (β = 0.269, T = 3.599, *p* = 0.000, with a small effect where f2 = 0.086), and social influence is positively affecting digital adoption (β = 0.225, T = 2.193, *p* = 0.014, with a small effect where f2 = 0.062). Lastly, the result shows that digital adoption positively affects SMEs' performance (β = 0.698, T value = 16.864, *p* value = 0.000, with a significant effect where f2 = 0.950), and SMEs performance positively affects SMEs sustainability (β = 0.765, T value = 21.714, *p* value = 0.000, with a significant effect where f2 = 1.411). Therefore, we can conclude that Hypotheses 8 and 9 were accepted. The result of hypotheses testing is described in the path analysis figure (see Figure 2).

**Table 5.** Path Coefficient.

| | β | T | *p* | Results | R2 | f2 | Q2 |
|---|---|---|---|---|---|---|---|
| Financial literacy → Financial Accessibility | 0.727 | 17.289 | 0.000 | Supported | 0.529 | 1.121 | 0.523 |
| Financial literacy → Financial Risk | 0.142 | 2.217 | 0.013 | Supported | 0.016 | 0.021 | 0.011 |
| Financial accessibility → digital adoption | 0.257 | 3.413 | 0.000 | Supported | | 0.073 | |
| Financial risk → digital adoption | 0.001 | 0.016 | 0.494 | Unsupported | | 0.000 | |
| Performance Expectancy → digital adoption | 0.154 | 1.936 | 0.026 | Supported | 0.571 | 0.024 | 0.554 |
| Effort Expectancy → digital adoption | 0.269 | 3.599 | 0.000 | Supported | | 0.086 | |
| Social influence → digital adoption | 0.225 | 2.193 | 0.014 | Supported | | 0.062 | |
| Digital adoption → SMEs performance | 0.698 | 16.864 | 0.000 | Supported | 0.485 | 0.950 | 0.450 |
| SMEs performance → SMEs sustainability | 0.765 | 21.714 | 0.000 | Supported | 0.583 | 1.411 | 0.368 |

The assessment of the structural model's predictive relevance (Q2) was the final phase of the structural model evaluation. As a result, we used the blindfolding technique recommended by Hair et al. [85]. According to Table 5, Q2 values greater than zero for both endogenous constructs of the model demonstrate sufficient predictive relevance.

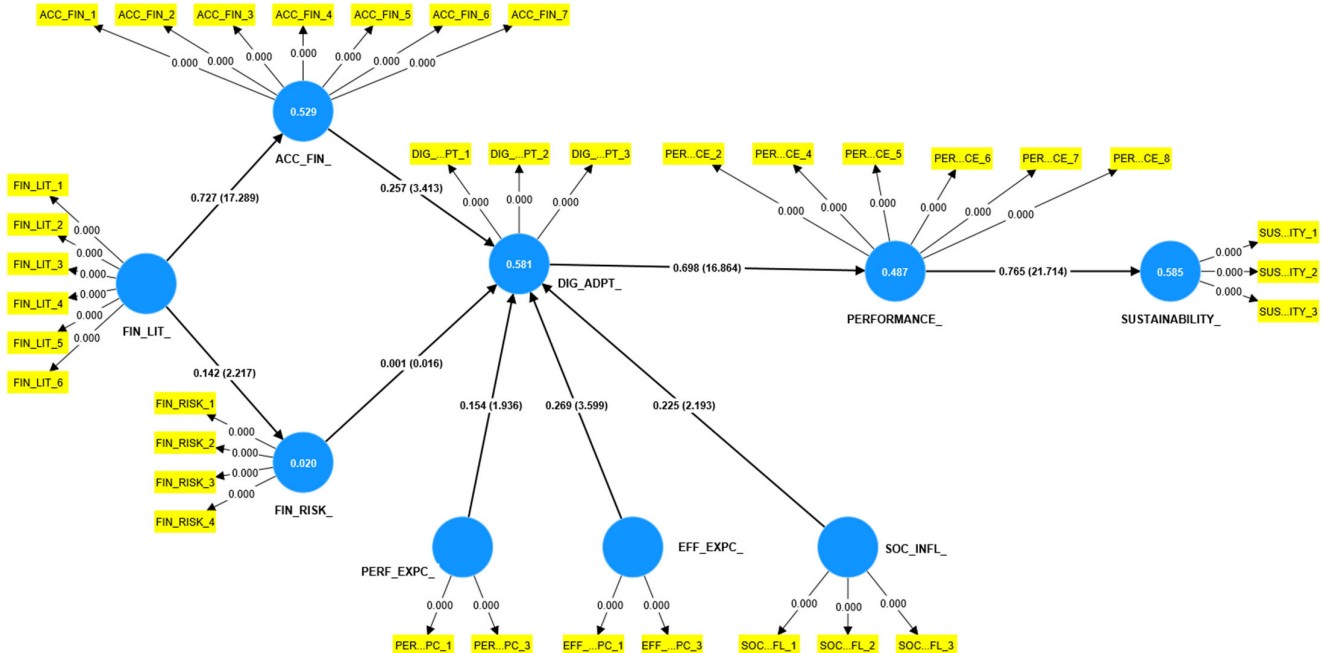

**Figure 2.** Result of Path Analysis.

## 5. Discussion

Traditional markets play a strategic role in the Indonesian economy because they are the largest contributor to government revenue from the retail sector. The number of traditional markets across Indonesia reached 13,450 units and could accommodate more than twelve million traders [12]. However, traditional marketplaces are becoming less and less popular among consumers, who now prefer to purchase in modern markets [13,14]. As the largest revenue contributor to the retail industry, the existence and continued existence of traditional markets hold a strategic role for the Indonesian government. Previous research on traditional market traders in Indonesia has revealed that the primary issues they confront involve financing [85,97]. From a financial standpoint, traditional market traders' lack of financial literacy has resulted in their continued reliance on loan sharks [98]. Most SMEs have difficulties when applying for loans to conventional financial institutions such as banks

because they are considered not bankable, even though from the business side it is already worthwhile or feasible. Despite this, the availability of FinTech peer-to-peer lending as an alternative financing option is still underutilized [99]. As a result, the study was carried out to investigate how SMEs financing factors, which include financial literacy, access to finance, and financial risk, as well as technological factors that consist of performance expectation, effort expectation, and social influence, affect the rate of technology adoption, business performance, and business sustainability in Indonesian SMEs and traditional market traders.

From the financial side, the results of this study show the positive impact of the level of financial literacy of managers on the financial accessibility or acceptance of Hypothesis 1 in this study. This confirms the results of the research conducted by Okello et al. [57] and Bustan et al. [100]. Financial literacy is a significant factor in an organization's intangible assets [2]. According to Okello et al. [57], financial literacy has a substantial impact on SMEs access to funding in developing nations. Greater access to financing for SMEs can help developing countries improve their economic conditions by encouraging innovation, macroeconomic resilience, and GDP growth [57]. Moreover, solid financial management is essential for the survival and administration of SMEs in emerging economies because financial literacy promotes awareness of numerous sources of funding, allowing SMEs to select and choose the best source of financing [100].

For Hypothesis 2, this study's results found that financial literacy positively affects SMEs financial risk. Our findings support prior findings on the effect of financial literacy on SME financial risk attitudes. In the process of making financial decisions, risk attitudes are influenced by cognitive and intuitive thinking patterns, which are influenced by financial literacy [101]. Entrepreneurs are better at risk management than others. Risk attitudes and risk management are determined by attitudes toward uncertainty and business conditions [102]. Ye and Kulathunga [2] found that SMEs with strong financial literacy are more likely to capitalize on highly profitable business prospects because they are ready to take risks.

The study found that only financial accessibility has a positive effect on digital adoption, while the effect of financial risk on digital adoption is insignificant. Therefore, Hypothesis 3 is supported, while Hypothesis 4 is not. The research implies that FinTech is expanding in areas where the present financial system is failing to meet demand for financial services [103]. Through the prudent deployment of technology, particularly the use of mobile phones to undertake financial services operations, marginalized and underserved groups gain access to previously unavailable financial resources [104]. In this example, the underserved groups that hardly access formal financial institutions are small to medium-sized businesses (SMEs). The establishment of FinTech can greatly aid the Indonesian government in achieving the country's long-term goals of boosting the digital economy, digital literacy, and financial inclusion in order to accelerate GDP growth [105]. Moreover, according to Frost [106], the existence and completeness of the FinTech information would affect customer decisions to use FinTech as an alternative solution for seeking business loans.

From a technology perspective, the study result confirms the positive effect of financial technology performance expectancy, social influence [61,71,72], and effort expectancy on SMEs digital adoption [73]. Therefore, Hypotheses 5, 6, and 7 are supported. According to Venkatesh et al. [45], technology adoption is strongly influenced by performance expectancy, effort expectancy, and social influence.

According to Venkatesh et al. [45], performance expectancy is described as the degree to which an individual believes that applying the system will aid in improving job performance. According to Najib, Ermawati, et al. [61] and Alshebami [71], performance expectancy positively affects financial technology adoption. Individuals might weigh the advantages and disadvantages of a system before deciding whether or not to utilize it [71]. The benefits of FinTech services and applications, such as the ability for individuals to con-

duct financial transactions from anywhere and at any time while using internet-connected devices, will drive FinTech adoption [72].

In terms of effort expectancy, customers will accept new technology if it is viewed as user-friendly and simple to use, allowing them to swiftly acquire it [61]. According to Chan et al. [73], making a financial innovation simple to use will also make it more beneficial and less hazardous to the users. According to Guild [107], the majority of SME owners in Indonesia are over forty years old. They are a less tech-savvy generation, so developing easy technology with simple navigation that is easy to learn will increase SMEs technology adoption. Moreover, in terms of the effect of social influence on technology adoption, this study confirms the importance of references, friends, families, social groups, and other individuals in making decisions about the usage of FinTech [61,71,72]. In other words, SMEs owners will probably adopt FinTech if other SMEs owners or their business partners already utilize it for their business purposes.

The study also shows the positive effect of digital adoption on SMEs performance (Hypothesis 8 is supported). Sjamsudin's [108] findings revealed that FinTech adoption has a positive impact on business sustainability performance. FinTech boosts the capacity of small firms, allowing them to be more competitive and thrive in the market [61]. Fin-Tech services such as digital payments, mobile investing platforms, and internet banking solutions have the ability to assist SMEs in improving their performance because FinTech has the capacity to increase the quality of products and services, which in turn improves SMEs operational and financial performance [76]. Moreover, FinTech speeds up the loan application procedure, allowing SMEs to access financing more quickly. Additionally, Fin-Tech may assist SMEs in obtaining lower-cost investment management advice. As a result, FinTech is expected to have an impact on the revenues and costs of SMEs [109]. Lastly, for Hypothesis 9, this study revealed that SMEs performance positively affects their business sustainability. According to Das et al. [110], digital innovation is critical to making small and medium-sized firms (SMEs) competitive and perform well in the market, which eventually makes for sustainable business growth. Moreover, according to Najib, Ermawati, et al. [61], the use of FinTech could improve the performance and sustainability of small enterprises. Small firms' capacity increases as a result of FinTech, allowing them to be more competitive and thrive in the market.

## 6. Conclusions

Traditional markets are an important part of the Indonesian economy because they are the main place where people buy and sell things. Despite this, the emergence of modern markets and the development of technology have caused the number of traders in traditional markets to decline. Given the important role played by traditional markets, this research is carried out to analyze the factors that influence the performance and sustainability of SMEs in the traditional market by looking at financial and technological factors.

From the financial side, the results confirm the positive impact of financial literacy as one of the intangible assets on access to finance and attitudes toward business risk. The findings also show a link between access to finance and financial technology adoption. From the technological side, the results of this study reinforce previous research findings that stated the positive influence of performance expectancy, effort expectancy, and social influence over financial technological adoption. Furthermore, the findings also confirm the positive impact of SME financial technology adoption on improved performance and business sustainability. Therefore, this study reinforces the pivotal role of financial technology in fostering financial inclusion. Financial technology has increased SMEs access to finance.

The findings of this study can be used by stakeholders such as the Ministry of Cooperation and SMEs, the Financial Services Authority, the FinTech industry, and traditional market traders. The emergence of FinTech as an alternative source of funding can thus be a solution to the financing challenges that continue to be the most significant impediments to the business performance and sustainability of SMEs in traditional markets. As a result,

the purpose of this study is to provide some insight. First, financial literacy as an essential component influencing SMEs capacity to acquire financial services needs to be improved by using financial terminology or concepts that are simple, brief, and not confusing. Second, in order to enhance FinTech acceptance, the Ministry of Cooperation and SMEs, in collaboration with the Financial Services Authority, can work with local governments and market managers to facilitate access for FinTech companies and educate SMEs in traditional markets about financial products. Third, governments should continue to monitor and analyze secure and trustworthy FinTech peer-to-peer lending on the financial risk front.

Although this study adds to the body of knowledge on SME digital adoption, SME performance, and business sustainability, it has significant shortcomings that may inspire further research. To begin, the sample was selected from only one city in Indonesia. Extending the sample to include the entire country would provide a more accurate picture. Second, the sample solely includes typical SMEs, which are traditional traders. Future research could also compare samples from modern market traders to uncover major elements that differentiate the existence of these two types of retail markets in the Indonesian environment. Third, future researchers are encouraged to conduct longitudinal studies to investigate the causal links between these variables.

**Author Contributions:** Following is the individual contribution to this research: Conceptualization: F.K. and E.D.L.; Research Methodology: F.K. and E.D.L.; Data Collection and Data Processing: H.T.; Writing—original draft preparation: F.K.; Writing—review and editing: F.K., E.D.L. and H.T.; Supervision: F.K.; Project Administration: E.D.L. All authors have read and agreed to the published version of the manuscript.

**Funding:** This research is funded by Universitas Multimedia Nusantara (0030-RD-LPPM-UMN/P-INT/II/2023) based on the Internal Research Contract for the year 2023.

**Institutional Review Board Statement:** This paper has pass Reasearch Ethic Clearance, with No. 0257-RD-LPPM/UMN/V/2023, signed by Director of LPPM and Head of Research UMN, on 16 May 2023.

**Informed Consent Statement:** Informed consent was obtained from all subjects involved in the study.

**Data Availability Statement:** Not applicable.

**Acknowledgments:** All authors give their highest appreciation to Universitas Multimedia Nusantara for their support in doing the research.

**Conflicts of Interest:** The authors declare no conflict of interest.

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
