# Peer review of "Pursuing Long-Term Business Performance: Investigating the Effects of Financial and Technological Factors on Digital Adoption to Leverage SME Performance and Business Sustainability—Evidence from Indonesian SMEs in the Traditional Market"

_sustainability, doi:10.3390/su151612668_

Round 1
Reviewer 1 Report (Previous Reviewer 2)
I mentioned before that text in tables should be left justified. Sure it's a matter of preference, but I really think it would look better left-justified. See, e.g., Table 2, 1st column.
I really think the Innovation and New Product Development book by Trott should be cited by you. As clarification: I'm not Trott, neither do I know him, nor am I related to him, yet I believe it's an important innovation text.
In order.
Author Response
Dear Reviewer,
We make some following changes:
- we adjust the table layout as suggested.
- we already cited Paul Trott and update the reference

Reviewer 2 Report (New Reviewer)
.

Author Response
we remove unrelated sentences in the context of low carbon economy

Round 2
Reviewer 2 Report (New Reviewer)
For reference 23, and others like that, the name of authors are indicated after by....who? and then will be the number.
Ussually, in the first sentence of Introduction is the claim of the paper, now is too generally formulated, be specific what the paper investigates and what contribution will have.
Also in Introduction it is not clear enough, so:
- Clearly state the specific research question or objective that the study aims to address
- and how it fills a gap in the existing knowledge.
Reformulate first sentence in line 410 indicating in the last part the Table 2...
Author Response
Dear Respective Reviewer,
We made some changes in the article based on your feedback. Some revisions include: adding the research questions, the explanation of theoretical gap as well as updated the citation

This manuscript is a resubmission of an earlier submission. The following is a list of the peer review reports and author responses from that submission.
Round 1
Reviewer 1 Report
This paper focuses on pursuing long-term business performance, investigating the effects of financial and technological factors on digital adoption, in order to leverage SME performance and business sus-tainability, through evidence from Indonesian SMEs in the Traditional Market.
The work is well is suitable for this journal.
However, in order to further improve it, I recommend to
1) Please highlight which is the novelty of your work.
2) Please highlight which are the main results.
3) Literature review could be further improved. In particular, I suggest considering the following work:
https://www.researchgate.net/publication/369147621_Sustainab-lization_Sustainability_and_Digitalization_as_a_Strategy_for_Resilience_in_the_Coffee_Sector
I encourage the authors to refine their paper to make it available for publication in the journal.
Author Response
Based on Reviewer 1 input, we made some revision, including:
- Highlight the novelty of the paper
- Highlight the main results
- Improving and updating the references to strengthen the literature review

Reviewer 2 Report
Title: Pursuing Long-Term Business Performance: Investigating the Effects of Financial and Technological Factors on Digital Adoption to Leverage SME Performance and Business Sustainability. Evidence from Indonesian SMEs in the Traditional Market
Introduction
Conceptually the article investigates the effects of financial and technological factors on the adoption of digital technologies to facilitate sustainability of SMEs in Indonesia. Nine hypotheses around financial and adoption aspects are defined and statistically tested through a research framework. The results indicate that some hypotheses are supported, but others not.
I have the following comments:
Comments
The title of the article is very long; I suggest you aim for 16 words max. In fact, from the title it’s hard to comprehend what the focus of the article is, so much so that as I was reading through the discussions on pages 2 and 3, I could not judge whether the said discussions are about the title (topic) of the article or not. For example, the pages contain a discussion on innovation, but the title does not mention it? Likewise, line 156 mentions “financial literacy”, but surely, the focus is wider than literacy?
Linking with the above, and further on in the “Discussion”, page 14 (lines 488 – 492): “Therefore, the research was conducted to look at how financial factors from the financial side, access to finance, and financial risk, as well as technological factors from performance expectation, effort expectation, and social influence, influence the rate of adoption of technology, business performance, and business sustainability of SMEs and traditional market traders in Indonesia.” – here the focus is again wider than financial literacy, even though it’s explicitly mentioned in Table 4.
Some rewording or further explanation of the financial literacy issue in the context of the wider financial issues are needed.
Abstract: Not clear in the Abstract what MSMEs stand for?
Lines 26 – 27: “While the effect of financial risk on 26 digital adoption is found to be insignificant.” Hard to comprehend as a sentence on its own? There are further instances too, e.g. “Aceh (31.25%) and East 89 Nusa Tenggara (20%).“ in lines 89 – 90. Kindly check all other occurrences of this kind.
Line 42, reference [2]: Maybe you can also look at and cite the Innovation Management and New Product Development book by Trott.
Lines 44 – 45 and elsewhere: Suggest you use “SMEs” or “an SME”. Then in line 213: Use “SMEs” instead of “SME’s”. Then, throughout (the rest of) the article, please check whether each context should be “SME”, “SMEs”, “SME’s”, etc.
Lines 78 – 80: Give a reference to the claim: “Based on the report released by We Are Social and KEPIOS, the number of Indonesians who access the internet continues to increase. 196.7 million users by 2020; 202.6 million users in 2021; and 210 million users in 2022).” Is this one in the list of references?
Line 152 (minor): Remove the yellow highlight. Line 154: Full stop missing at end of sentence. Also, add full stops at the end of the hypotheses sentences.
Sections 2.1 and 2.2: Discussions are about the resource-based view and technology acceptance/adoption, but the statistical model that you come up with later is mostly about financial issues with the exception of “Digital Adoption”. Should resource-based and technology aspects not feature more prominently in the model? Further, the citations in these sections are Harvard style, instead of the IEEE/ACM square bracket style. This may be indicative of text added afterwards. All these said, I take note of the text “The UTAUT variables (effort anticipation, performance expectancy, and social influence) in this study were mostly adapted from [37] and [58].” In lines 340 – 341, yet I believe a stronger link should be established, or at least more context or explanation in terms of the statistical model should be provided.
Line 388: Check also the Harvard citation in “(see in Hair, Sarstedt, et al., 2014).” Also “(Fornell & 389 Larcker, 1981)” in the next line.
Line 239: Orphan heading. Please check.
Line 318, reference [71]. I suspect this should be [72] instead? Please check the citation-reference mapping throughout.
Line 325: Large gap, but easily corrected.
Table 2: It’s generally a good idea to left-justify text in a table – see “Access to Finance” appearance. Also, repeat the header row of the table at the top of page 11. Same for tables 3, 4 each flowing over to the next page.
The figure between lines 435 – 436 should be given a caption. Further, since this is an important diagram, and the print is small, you may want to place the diagram on a landscape page of its own? And, of course, refer to it in the text.
Line 443: Incomplete sentence construct: “The greater the β value of the, the more …”.
End of line 586: “that may i …”. Hyphenation? Please check.
Conclusions section (page 16): The text reiterates that some hypotheses are supported, while others are not supported. Now, “standing back from the technicalities of the research”, what is/are the higher-level effect(s) of the support/non-support of the hypotheses? E.g., any managerial decisions that could be taken from these?
Finally, did you have to obtain ethical clearance to conduct the survey among humans? Even though it was a non-face-to-face survey?
There are some English gremlins - parts that are not full sentences, full stops missing, etc. But these are easily corrected.
Author Response
Based on Reviewer 2 input, we made some revision, including:
- Shortening the title
- We decided to use SMEs in the paper and made appropriate adjustment to ensure writing consistency
- We use the reference style ACS (American Chemical Society) that fit with the journal template
- Adding managerial implications for the findings of this research
- We did the ethical clearance and submit along with the revision
We already proofread our article to improve our English

Reviewer 3 Report
This is a good piece of work. I would certainly shorten the introduction and make it more coherent. It would also be advisable to clearly define "traditional market" as it is a vague term.
Decide whether you want to study MSMEs or SME, as they are certainly not the same. And shorten the title - it is too much descriptive and therefore far too long.
Revise the paper in order to improve the language.
Author Response
Based on Reviewer 3 input, we made some revision, including:
- We decided to use SMEs in the paper and made appropriate adjustment to ensure writing consistency
- We already proofread our article to improve our English

Reviewer 4 Report
Dear Authors,
I am highly delighted to review the article entitled Pursuing Long-Term Business Performance: Investigating the Effects of Financial and Technological Factors on Digital 3 Adoption to Leverage SME Performance and Business Sustainability." Evidence from Indonesian SMEs in the Traditional Market Though the work is interesting, there are many things that need updating in order to publish it.
1. The title is too big and needs to be concise.
2. Obviously, the abstract needs to be rewritten and made concise.
3. The articles are in good form, but I have to check the reliability of the data. Can you provide the data and screenshot of all your results and analyses during revision?
4. In the methodology part, I also need the variables names with the reference.
5. More justification is needed for why the authors chose the methodology.
6. The authors mentioned that respondent perceptions have been measured using a self-administered, validated 332 questionnaire. All items have been evaluated on a five-point Likert scale, with scores ranging from 1 (strongly disagree) to 5 (strongly agree). But from which sources have they identified that this study should follow this scoring to new findings?
7. How many months did it take you to collect the primary data?
8. Please cross-check all text citations with bibliographic citations.
Thank you
Good luck!
Author Response
Based on Reviewer 4 input, we made some revision, including:
- Shortening the title
- We provide the data of the results and analysis (please refer to the url link: https://drive.google.com/drive/folders/1KZRSaW3Y_auEulIKpB_FwJ5WnLAOO-xL?usp=sharing
- You can also access the online survey (in Bahasa Indonesia) through this link: Tingkat Adopsi Pembayaran Digital pada Pedagang Pasar (google.com)
- Adding the explanation about the methodology and Likert scale
- We need a month to collect the primary data since we had an assistance from the management of traditional market who had direct access to the sellers/traders
- We already cross check all the text citation

Reviewer 5 Report
Comments on the paper:
“Pursuing Long-Term Business Performance: Investigating the Effects of Financial and Technological Factors on Digital Adoption to Leverage SME Performance and Business Sustainability. Evidence from Indonesian SMEs in the Traditional Market”
Comments on the paper:
1. The English language in which the text is written needs extensive revision. See for example line 45 “99% are SME”, line 46 “absorbs 96.9% of national labor absorption”, lines 50-51 “but SME as the major victims”, line 60 “wholesale retail sales”, line 66 “studies that study”, line 187 “traders in develop their relationship”, line 189 “SME’s with strong financial literacy”, line 197 “Every SMEs should aware with”, line 242 “has a positively influence”, line 246 “performance expectancy positively affect”, line 271 “the level of others people effect”, line 282 “Several researchs demonstrating”.
2. In lines 39-40 the authors state “Sustainability not only includes sustainability for companies but also the responsibility of industry actors to the environment”. However, the authors do not specify in more detail how these two objectives are reconciled in the case of SMEs. The issue needs more elaboration.
3. In lines 45-46 the authors state that SMEs in Indonesia absorb 96.9% of national labour and contribute 60.5% to total GDP. These percentages seem too high to me. The public sector and large private sector enterprises almost hold negligible percentages of total employment?
4. What about the hidden or underground employment? How reliable are the official data about GDP and employment?
5. In lines 62-63 the authors state that “traditional markets experienced a negative growth of -8.1% while modern markets grew 31.4%”. For which reasons did these large deviations occurred?
6. The authors use as a general rule numbers for references. However, in some cases, e.g., in lines 163, 167, 171, 175 and 176, they refer directly to authors that are not included in the References section.
7. Financial accessibility is to a large extent “external” to SMEs, depending mainly to institutional factors. Namely to the institutional framework in which financial institutions operate. That framework would be useful to be described and explored in the context of this research.
8. In lines 422-424 the authors use a certain criterion proposed by a researcher to classify R2 values. Is that criterion generally acceptable? Do other researchers use that criterion?
9. I think that it would be very useful to examine some correlations between the variables. For example, whether the age, the educational level and the income of the enterprise significantly affect the results of empirical research.
10. In the section of Discussion the authors could provide some information about the limitations of their empirical research.
Comments on the paper:
“Pursuing Long-Term Business Performance: Investigating the Effects of Financial and Technological Factors on Digital Adoption to Leverage SME Performance and Business Sustainability. Evidence from Indonesian SMEs in the Traditional Market”
Comments on the paper:
1. The English language in which the text is written needs extensive revision. See for example line 45 “99% are SME”, line 46 “absorbs 96.9% of national labor absorption”, lines 50-51 “but SME as the major victims”, line 60 “wholesale retail sales”, line 66 “studies that study”, line 187 “traders in develop their relationship”, line 189 “SME’s with strong financial literacy”, line 197 “Every SMEs should aware with”, line 242 “has a positively influence”, line 246 “performance expectancy positively affect”, line 271 “the level of others people effect”, line 282 “Several researchs demonstrating”.
2. In lines 39-40 the authors state “Sustainability not only includes sustainability for companies but also the responsibility of industry actors to the environment”. However, the authors do not specify in more detail how these two objectives are reconciled in the case of SMEs. The issue needs more elaboration.
3. In lines 45-46 the authors state that SMEs in Indonesia absorb 96.9% of national labour and contribute 60.5% to total GDP. These percentages seem too high to me. The public sector and large private sector enterprises almost hold negligible percentages of total employment?
4. What about the hidden or underground employment? How reliable are the official data about GDP and employment?
5. In lines 62-63 the authors state that “traditional markets experienced a negative growth of -8.1% while modern markets grew 31.4%”. For which reasons did these large deviations occurred?
6. The authors use as a general rule numbers for references. However, in some cases, e.g., in lines 163, 167, 171, 175 and 176, they refer directly to authors that are not included in the References section.
7. Financial accessibility is to a large extent “external” to SMEs, depending mainly to institutional factors. Namely to the institutional framework in which financial institutions operate. That framework would be useful to be described and explored in the context of this research.
8. In lines 422-424 the authors use a certain criterion proposed by a researcher to classify R2 values. Is that criterion generally acceptable? Do other researchers use that criterion?
9. I think that it would be very useful to examine some correlations between the variables. For example, whether the age, the educational level and the income of the enterprise significantly affect the results of empirical research.
10. In the section of Discussion the authors could provide some information about the limitations of their empirical research.
Author Response
Based on Reviewer 5 input, we made some revision, including:
- Improving the language
- Adding limitation of the study

Round 2
Reviewer 5 Report
The authors took all my comments into consideration and revised their paper accordingly. I am satisfied with the revised version of their paper. I therefore propose the publication of the paper in its current form.
Sincerely yours,
Author Response
Thank you for your valuable comments and reviews.